# Effect of Nasal High Flow (NHF) on Right Heart Function in Stable Patients with Pulmonary Hypertension of Different WHO Classes

**DOI:** 10.3390/jcm13071862

**Published:** 2024-03-24

**Authors:** Jens Bräunlich, Hans-Jürgen Seyfarth, Hubert Wirtz

**Affiliations:** Department of Respiratory Medicine, University of Leipzig, 04103 Leipzig, Germany; hans-juergen.seyfarth@uniklinik-leipzig.de (H.-J.S.); hubert.wirtz@uniklinik-leipzig.de (H.W.)

**Keywords:** nasal high flow, high-flow nasal cannula, right hearth catheterization, pulmonary hypertension

## Abstract

**Background:** Nasal high flow (NHF) has various effects on the respiratory system in acute and chronic conditions. There are initial reports that NHF is also able to influence cardiac function in acute decompensation. This study was designed to clarify whether NHF has an influence on the right heart in stable patients with chronic pulmonary hypertension. **Methods:** Forty-one stable patients from different pulmonary hypertension (PH) WHO classes were recruited. Most patients were assigned to WHO classes 1 and 3. All received a right heart catheterization and blood gas analysis. Oxygenation was kept constant. The mean pulmonary arterial pressure (mPAP), wedge pressure (PC), cardiac output (CO), diastolic pulmonary gradient (DPG), pulmonary arterial resistance (PVR) and other parameters were determined. The patients then used NHF at 35 L/min for 20 min, after which the right heart catheter measurements were repeated with ongoing NHF therapy. **Results:** In the entire cohort and in the subgroups, there were no changes in right heart function or cardiac ejection fraction. The blood gases did not change either. **Conclusions:** Thus, there is no effect of NHF on right heart function in stable patients with PH.

## 1. Introduction

In recent years, nasal high flow (NHF) has developed into an important component of non-invasive respiratory support. The fact that it can be understood as more than just an oxygenation system plays a key role here. Other important physiological effects are the increased recruitment of lung areas, the increase in airway pressure, the washing out of the upper airways and the improved overcoming of dead space [1]. Clinically, these mechanisms lead to improved oxygenation, a decrease in hypercapnia and a reduction in the work of breathing. This results in an improved outcome in various clinical situations [2,3,4].

Not all areas of application for NHF have been sufficiently investigated. There are physiological effects that have so far only been investigated in case descriptions and smaller studies. These include humidification and warming of the airways. It is known that adequate humidification and warming of the airways leads to improved mucociliary activity [5]. This could be relevant for diseases that have an increased secretion load. One example of this is the recently published study by Crimi et al. This showed a significant reduction in the exacerbation rate and hospitalization and improvement in lung function [6]. A reduction in the exacerbation rate has also been shown for patients with COPD [4]. Another important, but still largely unexplored effect of NHF is its influence on cardiac function. Roca et al. provided initial data on this back in 2013, when they showed an improvement in echocardiographic parameters in acute cardiac decompensation [7]. Further clinical studies also showed positive effects in cardiac pulmonary edema (CPE) [8,9,10]. A recently published study retrospectively investigated the effects of NHF in patients with pulmonary hypertension (PH) in acute-on-chronic respiratory failure. Patients were re-evaluated by echocardiography after 10 days. There was an improvement in right heart morphology and a reduction in systolic pulmonary arterial pressure (sPAP) [11].

PH is defined by an increase in mean pulmonary arterial pressure (mPAP) greater than 20 mmHg. It is divided into different WHO groups. Two of the most common groups are groups 2 (PH due to left heart disease) and 3 (PH due to lung disease). A number of medications are used according to classification and risk [12,13]. Oxygen reduces pulmonary vascular resistance (PVR) in patients with pulmonary arterial hypertension. In WHO group 3 (PH due to lung disease), long-term oxygen therapy is often already in place in advanced stages of lung disease.

With the physiological effects already mentioned, NHF could therefore represent an opportunity to influence PH of various causes. Not only improved oxygenation could play a role here. Other mechanisms, such as the improvement of ventilation and the increase in airway pressure, could also play a role. This study was initiated to investigate this. In contrast to existing studies, right heart function was measured invasively in our study. The aim of this study was to find changes in right heart parameters during NHF use compared to baseline.

## 2. Methods

Subjects with an indication for a right heart catheterization, with suspected pulmonary hypertension, proven pulmonary hypertension for re-evaluation, or mandatory indication for right heart catheterization as part of a listing for lung transplantation for obstructive lung disease, older than 18 years were recruited. Suspicion was based on medical history and echocardiography. Subjects provided written informed consent to the trial protocol, including the analysis of their data, and were recruited from the respiratory ward at the university hospital of Leipzig from July 2016 to February 2018. This study was approved by the local ethics committee (461-15-24082015, approval date 12 July 2016) and registered (ClinicalTrails NCT02885116). The main exclusion criteria were acute respiratory insufficiency (pH < 7.35) with a lack of metabolic compensation, clinically unstable disease (e.g., acute myocardial infarction, severe exacerbation, cardiac arrhythmia, cardiac decompensation), serious concomitant diseases as assessed by the investigator, relevant systemic infections, simultaneous participation in other studies, lack of compliance/tolerance of therapy and women during pregnancy and lactation.

### 2.1. Device Use

In this study, we used a TNI softflow 50 device (TNI medical AG, Wuerzburg, Germany). NHF was applied using medium nasal prongs (TNI medical AG, Wuerzburg, Germany). After the first right heart catheter measurement was performed, the patient received NHF with a flow of 35 L/min for 20 min. The oxygen supply was adapted to achieve the same FIO2 and constant SpO2. The measurements of the right heart catheter were repeated during ongoing NHF therapy.

### 2.2. Hemodynamic Measurements

Cardiac catheterization was performed using standard hemodynamic methods [12]. Pulmonary artery pressure (PAP) was measured via jugularis interna using a fluid-filled 7.5 F catheter system. Curves were recorded for the right ventricular and pulmonary arterial pressure, as well as for the pulmonary capillary occlusion pressure. The central venous saturation was measured in the superior vena cava, in the right atrium and in the pulmonary artery with the aim of step oximetry. Cardiac output (CO) and the RVEF were measured by a thermodilution technique with an Explorer Eduard Life Sciences—System. PVR was estimated by the equation PVR = (mean PAP—pulmonary capillary wedge pressure [PCWP]) × CO^−1^ in SI units.

### 2.3. Capillary Blood Gas Measurement

Capillary blood gases were taken from the earlobe ten minutes after applying a hyperemia-inducing ointment (Finalgon^®^, Boehringer Ingelheim Pharma GmbH & Co., KG, Ingelheim am Rhein, Germany). Measurements were performed with constant oxygen flow before and after five hours of NHF use. The flow rate was chosen for maximal tolerability.

## 3. Statistics

Data were analyzed using ANOVA (Sigma Plot, Systat Software GmbH, version 7, Ekrath, Germany). A probability level for the null hypothesis (no difference) of 5% or below (*p* < 0.05) was accepted for significance. The results were expressed as mean values ± SD. Changes compared to baseline values were also expressed as percentages because of the variation in the baseline values. Measurements following ventilatory support were compared with baseline values. Significance was defined as a significant change from baseline.

## 4. Results

A total of 41 subjects were recruited. Of them, 34% were women, and 63% of the patients had long-term oxygen therapy. Most patients were classified as WHO class 3 (20 patients); 12 patients were classified as WHO class 1. Seven patients received PH medication (Table 1).

All included patients tolerated the procedure without side effects. There were no significant changes in the specific parameters under NHF therapy. A subgroup analysis, sorted according to WHO classes and separated into hypercapnia and normocapnia, also revealed no significant differences (Table 2).

## 5. Discussion

The aim of our study was to find differences in right heart function using NHF in patients with pulmonary hypertension. For this purpose, we examined patients in different WHO functional classes. All patients were in a stable disease phase. We were unable to find any differences in the parameters determined by right heart catheterization using NHF. This study is the first to look for an effect of NHF in chronically ill patients with pulmonary hypertension.

Our study forms a contrast to other studies that have addressed the question of the effect of NHF on the heart. The first study was published by Roca et al. It included 10 patients with WHO class 3 and heart failure. The outcome parameters were values measured by echocardiography. A greater collapse of the inferior vena cava (IVC) was found at 20 L/min (28%) and 40 L/min (37%). Other echocardiographic variables did not change. From this, the authors concluded that NHF is capable of being helpful in acute cardiac decompensation [7].

Further clinical studies confirmed this observation. Marjanovic et al. examined 27 patients with hypercapnic cardiac pulmonary edema. Briefly, 44% received NHF and 56% NIV. After 1 h, there was no difference between NIV and NHF in terms of respiratory parameters [9]. Ko et al. investigated the effect of NHF compared to oxygen in cardiac pulmonary edema. After allocation of the patients to the two groups, the 82 patients showed an improvement in respiratory parameters and blood gases. However, the ejection fraction was not improved [10]. Another study in this patient group also showed the effects of NHF, but also the superiority of the helmet continuous positive air pressure in comparison [8]. Unfortunately, no hemodynamic parameters were measured in these studies, so it is not clear whether the improvement under NHF was the result of a cardiac influence.

In the study by Pelaia et al., 17 patients with pulmonary hypertension and acute-to-chronic respiratory failure were examined. Conventional oxygen therapy (COT) using a Venturi mask was selected for comparison. This was compared with NHF with a flow rate of up to 60 L/min. The study mainly included patients with interstitial lung disease (ILD) or patients with obstructive pulmonary disease. The echocardiographic examinations showed an improvement in right ventricular function and a reduction in systolic pulmonary arterial pressure (sPAP) on day 10 compared to baseline. In detail, there was a decrease in the diameter of the IVC, the right atrium area (RAA) and the tricuspid annular systolic excursion (TAPSE). The left heart showed no changes. No differences in oxygenation were measured between the administration of COT and NHF [11].

There are various physiological explanations in the studies as to why the heart is relieved. On the one hand, the increase in lung volume through the recruitment of lung areas is discussed as one mechanism. This, together with the slight increase in intrathoracic pressure, could explain the lack of collapse of the inferior vena cava in the study by Roca et al. [7]. The study by Pelaia et al. also discusses an improvement in the ventilation–perfusion ratio, improvement in gas exchange and the administration of a humidified and heated gas mixture. All of these effects could play a role in the observed improvements and certainly have different effects in different patient groups [11]. It is known that in patients with PH, oxygen administration leads to an improvement in right ventricular parameters. Interestingly, however, in the studies by Pelaia et al. and Roca et al. there was no difference in oxygenation between COT and NHF [7,11]. This was also the case in our study, so that an improvement in oxygenation was not the main reason for the improvement in acute deterioration of the cardiac and respiratory situation.

Although our data do not show changes in the right heart, they do not exclude an effect on the right or left heart by NHF. The above studies included patients in an unstable phase of disease. In the study by Roca et al. this was heart failure and in the study by Pelaia et al. this was acute-to-chronic respiratory failure in the presence of underlying pulmonary disease [7,11]. Our study mainly involved patients with pulmonary hypertension due to underlying pulmonary disease or pulmonary arterial hypertension (PAH). Furthermore, all patients were in a stable disease phase. Another explanation may lie in the selected flow rate. While a flow rate of 35 L/min was used in our study, 20 and 40 L/min were also used in the study by Roca et al. Accordingly, these authors also found no changes in the parameters of the right heart, but only a stronger collapse of the inferior vena cava. In contrast, the study by Pelaia et al. used up to 60 L/min. This study also showed a decrease in the size of the inferior vena cava, but also in the RAA and the sPAP. Another difference of the studies is the duration of application of NHF. In the study by Roca et al., NHF was used for 60 min; in the study by Pelaia et al., it was used for 10 days. This makes the application of NHF in our study the shortest duration at 20 min. Whether the time alone has an influence on the result is unclear. After all, the patient groups are too different between the studies to be able to identify this as the sole factor.

However, it is not only the sole application of NHF that can have an effect on the right heart. The application of medication via NHF has been an important application for some years now [14]. In addition to bronchodilators, drugs were also tested. An interesting study was published by Li et al. Here, epoprostenol, a drug to reduce pulmonary pressure, was applied. Titration of the flow rate (together with epoprostenol administration) resulted in a greater improvement in right heart function than the administration of constant flow [15].

It is known that non-invasive respiratory support also has negative cardiac effects. The use of low- and high-intensity non-invasive ventilation (NIV) was investigated by Lukacsovits et al. Although high-intensity NIV showed an improvement in gas exchange, cardiac output deteriorated [16]. This has already been observed in other studies, even with lower pressure support [17,18]. Such an effect is not to be expected when using NHF. This is mainly due to the fact that NHF is not a closed system. Accordingly, the pressure in the airways cannot increase as much as with NIV. This means that negative effects of NHF on the heart can be largely ruled out. This is clearly shown by the studies available to date on this topic [7,11].

We also conducted subgroup analyses in our study (patients with PAH, patients with lung disease, hypercapnic patients). None of the groups showed a change in the parameters under NHF therapy.

Our study has limitations. Only patients in a stable disease phase were measured. This does not rule out the possibility that NHF may be effective in the event of decompensation of cardiac and/or respiratory function. Furthermore, NHF was only applied for 20 min. Furthermore, a flow rate was used (35 L/min) which is below the flow rate used in acute respiratory failure (up to 60 L/min). We decided to use these settings because the best compliance in stable patients is at 30–35 L/min. Likewise, this elective examination was only tolerated by the patients over a certain period of time. The data situation at the beginning of this study and also today was hardly helpful in determining the optimal parameters. It is therefore quite possible that prolonged use of NHF with a higher flow rate could have effects on the heart. In addition, we did not recruit any patients in PH class 2 (PH due to left heart disease). We did not deliberately exclude patients in group 2, but none of these patients had an indication for a right heart catheterization during the study period.

## 6. Conclusions

In our study in stable patients with pulmonary hypertension of different WHO classes, we found no differences in the use of NHF compared to baseline in right heart catheterization. This does not exclude the possibility that an effect can be achieved in other patient groups and by using higher flow rates and a longer utilization time.

## Figures and Tables

**Table 1 jcm-13-01862-t001:** Demographic data. *n* = 41, PH = pulmonary hypertension, SaO2 = arterial saturation of oxygen, paO2 = arterial partial pressure of oxygen, paCO2 = arterial partial pressure of carbon dioxide.

Age in years	61.5 ± 9.8
Women/men	34%/66%
Height in cm	172 ± 9.6
Weight in kg	77 ± 20.5
SaO2 in %	93.2 ± 4.5
paO2 in mmHg	70.9 ± 13.9
paCO2 in mmHg	40.4 ± 12.2
Long-term oxygen therapy	26 patients
Oxygen supplementation L/min	1.5 ± 1.4
PH WHO group 1	12
PH WHO group 2	0
PH WHO group 3	20
PH WHO group 4	6
PH WHO group 5	3
Specific therapy	3 patients Sildenafil, 2 patients Tadalafil, 1 patient SildenafiL/Bosentan, 1 patient Bosentan/SildenafiL/Prostacyclin

**Table 2 jcm-13-01862-t002:** Results of right heart catheterization at baseline and after use of the NHF.

	Baseline	NHF 35 L/min	*p*-Values
PAPm (mmHg)	41.5 ± 13.9	40.3 ± 13.6	0.68
PC (mmHg)	13.0 ± 12.5	13.2 ± 13.9	0.95
CO (L/min)	4.9 ± 1.5	4.9 ± 1.4	0.97
CI (L/min/qm)	2.8 ± 0.7	2.6 ± 0.6	0.35
ZVD (mmHg)	9.5 ± 5.9	9.1 ± 6.0	0.80
DPG (mmHg)	17.9 ± 9.3	16.3 ± 9.1	0.46
SaO2 %	93.9 ± 3.4	93.2 ± 4.5	0.46
paO2 (mmHg)	70.9 ± 13.9	67.1 ± 14.9	0.23
paCO2 (mmHg)	40.4 ± 12.2	38.6 ± 11.6	0.49
SvO2 %	68.1 ± 8.5	64.4 ± 9.9	0.07
RVEF %	29.6 ± 11.1	30.2 ± 12.1	0.80
PVR (dyn·s·cm^−5^)	555.9 ± 465.7	544.4 ± 465.8	0.95
SVI (mL/qm)	37.5 ± 12.9	36.9 ± 13.2	0.85
DO2 (mL/min)	1279.9 ± 608.8	1077.1 ± 458.7	0.16
VO2 (mL/min)	359.5 ± 157.8	340.9 ± 125.0	0.62
VO2/DO2 (%)	29.6 ± 8.3	32.9 ± 7.0	0.12

PAPm = mean pulmonary arterial pressure, PC = wedge pressure, CO = cardiac output, CI = cardi-ac index, ZVD = central venous pressure, DPG = diastolic pulmonary gradient, SaO2 = arterial sat-uration of oxygen, paO2 = arterial partial pressure of oxygen, paCO2 = arterial partial pressure of carbon dioxide, SvO2 = venous oxygen saturation , RVEF = right ventricular ejection fraction, PVR = pulmonary arterial resistance, SVI = stroke volume index, DO2 = oxygen delivery, VO2 = oxygen consumption, VO2/DO2 = oxygen extraction ratio. p-values > 0.05 indicate a non-significant differ-ence.

## Data Availability

Data are available upon request to the corresponding author.

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
