# Peer review of "Effect of Nasal High Flow (NHF) on Right Heart Function in Stable Patients with Pulmonary Hypertension of Different WHO Classes"

_jcm, 2024, doi:10.3390/jcm13071862_

Round 1

Reviewer 1 Report

Comments and Suggestions for Authors

I am really appreciate to giving me possibilitty to review this article. 

On the begining I would notice that I am really satisfied that new investigations on HFNC appears, especially concerning heart.

In my opinion this research could be better design - the flow 35l/min is too low and didn't give real effect of using HFNC, what also authors noticed and metioned in the discussion section. Why the authors didn't give higher parameters of ventilation in designed study, despite they mentioned that higher flow had given better effect in previous investigations? 

Additional I would like to ask why protocol of investigation assumed only 20 minutes of ventilation - in my opinion it's too short time to give real effect of using HFNC, especially where patients are stable.

Moreover there were used different groups of patient with PH, but why authors didn't include patients with pulmonary hypertension due to left heart diseases?

I think it is worth briefly mentioning in the summary that the use of HFNC did not bring any results, but this could have been due to too low parameters and ventilation time.

Reviewer 2 Report

Comments and Suggestions for Authors

Manuscript ID: 2905679

Manuscript title: Effect of nasal high flow (NHF) on right heart function in stable patients with pulmonary hypertension of different WHO classes

Line no 2. Authors mentioned manuscript title.. please remove

Abstract: Less informative. Kindly include more results into the abstract.

Line no 60, 61, 62 ,66,67 and 68 under methods authors please clearly mention the inclusion criteria and exclusion criteria of the study clearly.

Table 1 authors mentioned woman 34% what about men participants… are they excluded? Why? Clarify?

Authors please check the abbreviation for patients ..you mentioned Pat in the table 1.. I am not sure whether it is correct. If you don’t find mention as patients.

Table 2  authors mentioned >0,05…it means you are trying to say not significant? Explain and correct it.

Please mention the actual p value

Please change

Table 1 and Table 2 everywhere it has, symbol it must be. Between the decimal.

Table 2  Column row 7…saO2 in that s is small please check and make s capital.

Authors please mention the separate conclusion section next to the Discussion

It must have some valid strong information

Good luck

Comments on the Quality of English Language

Minor english language editing is required

Round 2

Reviewer 1 Report

Comments and Suggestions for Authors

Thank You for your response, I think that your describe it in appropriate way. 

With best regards